# Learning Reward Machines for Partially Observable Reinforcement Learning

**Rodrigo Toro Icarte**[*]
University of Toronto
Vector Institute

**Ethan Waldie**
University of Toronto

**Toryn Q. Klassen**
University of Toronto
Vector Institute

**Richard Valenzano**
Element AI

**Margarita P. Castro**
University of Toronto

**Sheila A. McIlraith**
University of Toronto
Vector Institute

## Abstract

Reward Machines (RMs) provide a structured, automata-based representation of a reward function that enables a Reinforcement Learning (RL) agent to decompose an RL problem into structured subproblems that can be efficiently learned via off-policy learning. Here we show that RMs can be learned from experience, instead of being specified by the user, and that the resulting problem decomposition can be used to effectively solve partially observable RL problems. We pose the task of learning RMs as a discrete optimization problem where the objective is to find an RM that decomposes the problem into a set of subproblems such that the combination of their optimal memoryless policies is an optimal policy for the original problem. We show the effectiveness of this approach on three partially observable domains, where it significantly outperforms A3C, PPO, and ACER, and discuss its advantages, limitations, and broader potential.[1]

## 1 Introduction

The use of neural networks for function approximation has led to many recent advances in *Reinforcement Learning (RL)*. Such deep RL methods have allowed agents to learn effective policies in many complex environment including board games [30], video games [23], and robotic systems [2]. However, RL methods (including deep RL methods) often struggle when the environment is *partially observable*. This is because agents in such environments usually require some form of memory to learn optimal behaviour [31]. Recent approaches for giving memory to an RL agent either rely on recurrent neural networks [24, 15, 37, 29] or memory-augmented neural networks [25, 18].

In this work, we show that *Reward Machines (RMs)* are another useful tool for providing memory in a partially observable environment. RMs were originally conceived to provide a structured, automata-based representation of a reward function [33, 4, 14, 39]. Exposed structure can be exploited by the *Q-Learning for Reward Machines (QRM)* algorithm [33], which simultaneously learns a separate policy for each state in the RM. QRM has been shown to outperform standard and hierarchical deep RL over a variety of discrete and continuous domains. However, QRM was only defined for fully observable environments. Furthermore, the RMs were handcrafted.

In this paper, we propose a method for learning an RM directly from experience in a partially observable environment, in a manner that allows the RM to serve as memory for an RL algorithm.

---

[*]Correspondence to: Rodrigo Toro Icarte <rntoro@cs.toronto.edu>.
[1]Our code is available at https://bitbucket.org/RToroIcarte/lrm.

A requirement is that the RM learning method be given a finite set of detectors for properties that serve as the vocabulary for the RM. We characterize an objective for RM learning that allows us to formulate the task as a discrete optimization problem and propose an efficient local search approach to solve it. By simultaneously learning an RM and a policy for the environment, we are able to significantly outperform several deep RL baselines that use recurrent neural networks as memory in three partially observable domains. We also extend QRM to the case of partial observability where we see further gains when combined with our RM learning method.

## 2 Preliminaries

RL agents learn policies from experience. When the problem is fully-observable, the underlying environment model is typically assumed to be a *Markov Decision Process (MDP)*. An MDP is a tuple $\mathcal{M} = \langle S, A, r, p, \gamma \rangle$, where $S$ is a finite set of *states*, $A$ is a finite set of *actions*, $r : S \times A \to \mathbb{R}$ is the *reward function*, $p(s, a, s')$ is the *transition probability distribution*, and $\gamma$ is the *discount factor*. The agent starts not knowing what $r$ or $p$ are. At every time step $t$, the agent observes the current state $s_t \in S$ and executes an action $a_t \in A$ following a policy $\pi(a_t|s_t)$. As a result, the state $s_t$ changes to $s_{t+1} \sim p(s_{t+1}|s_t, a_t)$ and the agent receives a reward signal $r(s_t, a_t)$. The goal is to learn the *optimal* policy $\pi^*$, which maximizes the future expected discounted reward for every state in $S$ [32].

*Q-learning* [38] is a well-known RL algorithm that uses samples of experience of the form $(s_t, a_t, r_t, s_{t+1})$ to estimate the optimal q-function $q^*(s, a)$. Here, $q^*(s, a)$ is the expected return of selecting action $a$ in state $s$ and following an optimal policy $\pi^*$. Deep RL methods like DQN [23] and DDQN [35] represent the q-function as $\tilde{q}_\theta(s, a)$, where $\tilde{q}_\theta$ is a neural network whose inputs are features of the state and action, and whose weights $\theta$ are updated using stochastic gradient descent.

In partially observable problems, the underlying environment model is typically assumed to be a *Partially Observable Markov Decision Process (POMDP)*. A POMDP is a tuple $\mathcal{P}_{\mathcal{O}} = \langle S, O, A, r, p, \omega, \gamma \rangle$, where $S$, $A$, $r$, $p$, and $\gamma$ are defined as in an MDP, $O$ is a finite set of *observations*, and $\omega(s, o)$ is the *observation probability distribution*. At every time step $t$, the agent is in exactly one state $s_t \in S$, executes an action $a_t \in A$, receives reward $r_t = r(s_t, a_t)$, and moves to state $s_{t+1}$ according to $p(s_t, a_t, s_{t+1})$. However, the agent does not observe $s_{t+1}$, but only receives an observation $o_{t+1} \in O$. This observation provides the agent a clue about what the state $s_{t+1} \in S$ is via $\omega$. In particular, $\omega(s_{t+1}, o_{t+1})$ is the probability of observing $o_{t+1}$ from state $s_{t+1}$ [5].

RL methods cannot be immediately applied to POMDPs because the transition probabilities and reward function are not necessarily Markovian w.r.t. $O$ (though by definition they are w.r.t. $S$). As such, optimal policies may need to consider the complete history $o_0, a_0, \ldots, a_{t-1}, o_t$ of observations and actions when selecting the next action. Several partially observable RL methods use a recurrent neural network to compactly represent the history, and then use a policy gradient method to train it. However, when we do have access to a full POMDP model $\mathcal{P}_{\mathcal{O}}$, then the history can be summarized into a *belief state*. A belief state is a probability distribution $b_t : S \to [0, 1]$ over $S$, such that $b_t(s)$ is the probability that the agent is in state $s \in S$ given the history up to time $t$. The initial belief state is computed using the initial observation $o_0$: $b_0(s) \propto \omega(s, o_0)$ for all $s \in S$. The belief state $b_{t+1}$ is then determined from the previous belief state $b_t$, the executed action $a_t$, and the resulting observation $o_{t+1}$ as $b_{t+1}(s') \propto \omega(s', o_{t+1}) \sum_{s \in S} p(s, a_t, s') b_t(s)$ for all $s' \in S$. Since the state transitions and reward function are Markovian w.r.t. $b_t$, the set of all belief states $B$ can be used to construct the belief MDP $\mathcal{M}_B$. Optimal policies for $\mathcal{M}_B$ are also optimal for the POMDP [5].

## 3 Reward Machines for Partially Observable Environments

In this section, we define RMs for the case of partial observability. We use the following problem as a running example to help explain various concepts.

**Example 3.1** (The cookie domain). *The* cookie domain*, shown in Figure 1a, has three rooms connected by a hallway. The agent (purple triangle) can move in the four cardinal directions. There is a button in the yellow room that, when pressed, causes a cookie to randomly appear in the red or blue room. The agent receives a reward of +1 for reaching (and thus eating) the cookie and may then go and press the button again. Pressing the button before reaching a cookie will move it to a random location. There is no cookie at the beginning of the episode. This is a partially observable environment since the agent can only see what it is in the room that it currently occupies.*

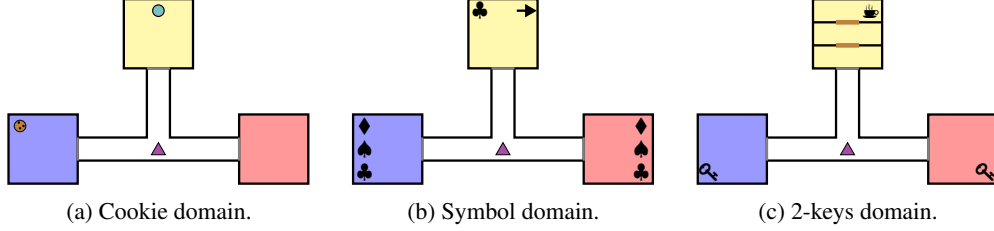

<div align="center">(a) Cookie domain.      (b) Symbol domain.      (c) 2-keys domain.</div>

Figure 1: Partially observable environments where the agent can only see what is in the current room.

RMs are finite state machines that are used to encode a reward function [33]. In the case of partial observability, they are defined over a set of propositional symbols $\mathcal{P}$ that correspond to a set of high-level features that the agent can detect using a *labelling function* $L : O_\emptyset \times A_\emptyset \times O \to 2^{\mathcal{P}}$ where (for any set $X$) $X_\emptyset \triangleq X \cup \{\emptyset\}$. $L$ assigns truth values to symbols in $\mathcal{P}$ given an environment experience $e = (o, a, o')$ where $o'$ is the observation seen after executing action $a$ when observing $o$. We use $L(\emptyset, \emptyset, o)$ to assign truth values to the initial observation. We call a truth value assignment of $\mathcal{P}$ an *abstract observation* because it provides a high-level view of the low-level environment observations via the labelling function $L$. A formal definition of an RM follows:

**Definition 3.1** (reward machine). Given a set of propositional symbols $\mathcal{P}$, a Reward Machine is a tuple $\mathcal{R}_\mathcal{P} = \langle U, u_0, \delta_u, \delta_r \rangle$ where $U$ is a finite set of states, $u_0 \in U$ is an initial state, $\delta_u$ is the state-transition function, $\delta_u : U \times 2^{\mathcal{P}} \to U$, and $\delta_r$ is the reward-transition function, $\delta_r : U \times 2^{\mathcal{P}} \to \mathbb{R}$.

RMs decompose problems into a set of high-level states $U$ and define transitions using if-like conditions defined by $\delta_u$. These conditions are over a set of binary properties $\mathcal{P}$ that the agent can detect using $L$. For example, in the cookie domain, $\mathcal{P} = \{$🍪, ☺, ◯, 🟥, ☐, 🟦, 🟨$\}$. These properties are true (i.e., part of an experience label according to $L$) in the following situations: 🟥, ☐, 🟦, or 🟨 is true if the agent ends the experience in a room of that color; 🍪 is true if the agent ends the experience in the same room as a cookie; ◯ is true if the agent pushed the button with its last action; and ☺ is true if the agent ate a cookie with its last action (by moving onto the space where the cookie was).

Figure 2 shows three possible RMs for the cookie domain. They all define the same reward signal (1 for eating a cookie and 0 otherwise) but differ in their states and transitions. As a result, they differ with respect to the amount of information about the current domain state that can be inferred from the current RM state, as we will see below.

Each RM starts in the initial state $u_0$. Edge labels in the figures provide a visual representation of the functions $\delta_u$ and $\delta_r$. For example, label $\langle$🟥☺$, 1\rangle$ between state $u_2$ and $u_0$ in Figure 2b represents $\delta_u(u_2, \{$🟥$,$☺$\}) = u_0$ and $\delta_r(u_2, \{$🟥$,$☺$\}) = 1$. Intuitively, this means that if the RM is in state $u_2$ and the agent's experience ended in room 🟥 immediately after eating the cookie ☺, then the agent will receive a reward of 1 and the RM will transition to $u_0$. Notice that any properties not listed in the label are false (e.g. 🍪 must be false to take the transition labelled $\langle$🟥☺$, 1\rangle$). We also use multiple labels separated by a semicolon (e.g., "$\langle$🟦$, 0\rangle; \langle$🟥🍪$, 0\rangle$") to describe different conditions for transitioning between the RM states, each with their own associated reward. The label $\langle$o/w$, r\rangle$ ("o/w" for "otherwise") on an edge from $u_i$ to $u_j$ means that that transition will be made (and reward $r$ received) if none of the other transitions from $u_i$ can be taken.

Let us illustrate the behaviour of an RM using the one shown in Figure 2c. The RM will stay in $u_0$ until the agent presses the button (causing a cookie to appear), whereupon the RM moves to $u_1$. From $u_1$ the RM may move to $u_2$ or $u_3$ depending on whether the agent finds a cookie when it enters another room. It is also possible to associate meaning with being in RM states: $u_0$ means that there is no cookie available, $u_1$ means that there is a cookie in some room (either blue or red), etc.

When learning a policy for a given RM, one simple technique is to learn a policy $\pi(o, u)$ that considers the current observation $o \in O$ and the current RM state $u \in U$. Interestingly, a partially observable problem might be non-Markovian over $O$, but Markovian over $O \times U$ for some RM $\mathcal{R}_\mathcal{P}$. This is the case for the cookie domain with the RM from Figure 2c, for example.

*Q-Learning for RMs (QRM)* is another way to learn a policy by exploiting a given RM [33]. QRM learns one q-function $\tilde{q}_u$ (i.e., policy) per RM state $u \in U$. Then, given any sample experience, the RM can be used to emulate how much reward would have been received had the RM been in

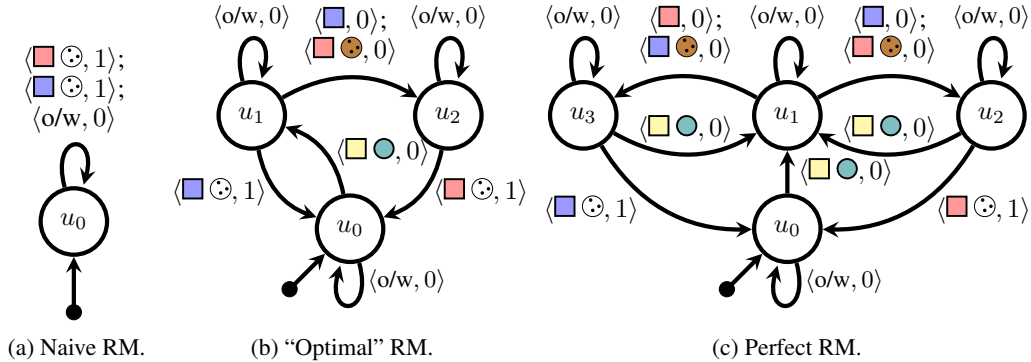

<div align="center">

(a) Naive RM.      (b) "Optimal" RM.      (c) Perfect RM.

Figure 2: Three possible Reward Machines for the Cookie domain.

</div>

any one of its states. Formally, experience $e = (o, a, o')$ can be transformed into a valid experience $(\langle o, u \rangle, a, \langle o', u' \rangle, r)$ used for updating $\tilde{q}_u$ for each $u \in U$, where $u' = \delta_u(u, L(e))$ and $r = \delta_r(u, L(e))$. Hence, any off-policy learning method can take advantage of these "synthetically" generated experiences to update all subpolicies simultaneously.

When tabular q-learning is used, QRM is guaranteed to converge to an optimal policy on fully-observable problems [33]. However, in a partially observable environment, an experience $e$ might be more or less likely depending on the RM state that the agent was in when the experience was collected. For example, experience $e$ might be possible in one RM state $u_i$ but not in RM state $u_j$. Thus, updating the policy for $u_j$ using $e$ as QRM does, would introduce an unwanted bias to $\tilde{q}_{u_j}$. We will discuss how to (partially) address this problem in §5.

## 4 Learning Reward Machines from Traces

Our overall idea is to search for an RM that can be used as external memory by an agent for a given task. As input, our method will only take a set of high-level propositional symbols $\mathcal{P}$, and a labelling function $L$ that can detect them. Then, the key question is what properties should such an RM have.

Three proposals naturally emerge from the literature. The first comes from the work on learning Finite State Machines (FSMs) [3, 40, 10], which suggests learning the smallest RM that correctly mimics the external reward signal given by the environment, as in Giantamidis and Tripakis' method for learning Moore Machines [10]. Unfortunately, such approaches would learn RMs of limited utility, like the one in Figure 2a. This naive RM correctly predicts reward in the cookie domain (i.e., +1 for eating a cookie ☺, zero otherwise) but provides no memory in support of solving the task.

The second proposal comes from the literature on learning Finite State Controllers (FSC) [22] and on model-free RL methods [32]. This work suggests looking for the RM whose optimal policy receives the most reward. For instance, the RM from Figure 2b is "optimal" in this sense. It decomposes the problem into three states. The optimal policy for $u_0$ goes directly to press the button, the optimal policy for $u_1$ goes to the blue room and eats the cookie if present, and the optimal policy for $u_2$ goes to the red room and eats the cookie. Together, these three policies give rise to an optimal policy for the complete problem. This is a desirable property for RMs, but requires computing optimal policies in order to compare the relative quality of RMs, which seems prohibitively expensive. However, we believe that finding ways to efficiently learn "optimal" RMs is a promising future work direction.

Finally, the third proposal comes from the literature on Predictive State Representations (PSR) [20], Deterministic Markov Models (DMMs) [21], and model-based RL [16]. These works suggest learning the RM that remembers sufficient information about the history to make accurate Markovian predictions about the next observation. For instance, the cookie domain RM shown in Figure 2c is *perfect* w.r.t. this criterion. Intuitively, every transition in the cookie environment is already Markovian except for transitioning from one room to another. Depending on different factors, when entering to the red room there could be a cookie there (or not). The perfect RM is able to encode such information using 4 states: when at $u_0$ the agent knows that there is no cookie, at $u_1$ the agent knows that there is a cookie in the blue or the red room, at $u_2$ the agent knows that there is a cookie

in the red room, and at $u_3$ the agent knows that there is a cookie in the blue room. Since keeping track of more information will not result in better predictions, this RM is *perfect*. Below, we develop a theory about perfect RMs and describe an approach to learn them.

## 4.1 Perfect Reward Machines: Formal Definition and Properties

The key insight behind perfect RMs is to use their states $U$ and transitions $\delta_u$ to keep track of relevant past information such that the partially observable environment $\mathcal{P}_{\mathcal{O}}$ becomes Markovian w.r.t. $O \times U$.

**Definition 4.1** (perfect reward machine). An RM $\mathcal{R}_{\mathcal{P}} = \langle U, u_0, \delta_u, \delta_r \rangle$ is considered perfect for a POMDP $\mathcal{P}_{\mathcal{O}} = \langle S, O, A, r, p, \omega, \gamma \rangle$ with respect to a labelling function $L$ if and only if for every trace $o_0, a_0, \ldots, o_t, a_t$ generated by any policy over $\mathcal{P}_{\mathcal{O}}$, the following holds:

$$\Pr(o_{t+1}, r_t | o_0, a_0, \ldots, o_t, a_t) = \Pr(o_{t+1}, r_t | o_t, x_t, a_t) \tag{1}$$

where $x_0 = u_0$ and $x_t = \delta_u(x_{t-1}, L(o_{t-1}, a_{t-1}, o_t))$ .

Two interesting properties follow from Definition 4.1. First, if the set of belief states $B$ for the POMDP $\mathcal{P}_{\mathcal{O}}$ is finite, then there exists a perfect RM for $\mathcal{P}_{\mathcal{O}}$ with respect to some $L$. Second, the optimal policies for perfect RMs are also optimal for the POMDP (see supplementary material §3).

**Theorem 4.1.** *Given any POMDP $\mathcal{P}_{\mathcal{O}}$ with a finite reachable belief space, there will always exists at least one perfect RM for $\mathcal{P}_{\mathcal{O}}$ with respect to some labelling function $L$.*

**Theorem 4.2.** *Let $\mathcal{R}_{\mathcal{P}}$ be a perfect RM for a POMDP $\mathcal{P}_{\mathcal{O}}$ w.r.t. a labelling function $L$, then any optimal policy for $\mathcal{R}_{\mathcal{P}}$ w.r.t. the environmental reward is also optimal for $\mathcal{P}_{\mathcal{O}}$.*

## 4.2 Perfect Reward Machines: How to Learn Them

We now consider the problem of learning a perfect RM from traces, assuming one exists w.r.t. the given labelling function $L$. Recall that a perfect RM transforms the original problem into a Markovian problem over $O \times U$. Hence, we should prefer RMs that accurately predict the next observation $o'$ and immediate reward $r$ from the current observation $o$, RM state $u$, and action $a$. This might be achieved by collecting a training set of traces from the environment, fitting a predictive model for $\Pr(o', r | o, u, a)$, and picking the RM that makes better predictions. However, this can be very expensive, especially considering that the observations might be images.

Instead, we propose an alternative that focuses on a necessary condition for a perfect RM: the RM must predict what is *possible* and *impossible* in the environment at the abstract level. For example, it is impossible to be at $u_3$ in the RM from Figure 2c and make the abstract observation $\{\blacksquare, \bullet\}$, because the RM reaches $u_3$ only if the cookie was seen in the blue room or not to be in the red room.

This idea is formalized in the optimization model LRM. Let $\mathcal{T} = \{\mathcal{T}_0, \ldots, \mathcal{T}_n\}$ be a set of traces, where each trace $\mathcal{T}_i$ is a sequence of observations, actions, and rewards:

$$\mathcal{T}_i = (o_{i,0}, a_{i,0}, r_{i,0}, \ldots, a_{i,t_i-1}, r_{i,t_i-1}, o_{i,t_i}). \tag{2}$$

We now look for an RM $\langle U, u_0, \delta_u, \delta_r \rangle$ that can be used to predict $L(e_{i,t+1})$ from $L(e_{i,t})$ and the current RM state $x_{i,t}$, where $e_{i,t+1}$ is the experience $(o_{i,t}, a_{i,t}, o_{i,t+1})$ and $e_{i,0}$ is $(\emptyset, \emptyset, o_{i,0})$ by definition. The model parameters are the set of traces $\mathcal{T}$, the set of propositional symbols $\mathcal{P}$, the labelling function $L$, and a maximum number of states in the RM $u_{\max}$. The model also uses the sets $I = \{0 \ldots n\}$ and $T_i = \{0 \ldots t_i - 1\}$, where $I$ contains the index of the traces and $T_i$ their time steps. The model has two auxiliary variables $x_{i,t}$ and $N_{u,l}$. Variable $x_{i,t} \in U$ represents the state of the RM after observing trace $\mathcal{T}_i$ up to time $t$. Variable $N_{u,l} \subseteq 2^{2^{\mathcal{P}}}$ is the set of all the next abstract observations seen from the RM state $u$ and the abstract observations $l$ at some point in $\mathcal{T}$. In other words, $l' \in N_{u,l}$ iff $u = x_{i,t}$, $l = L(e_{i,t})$, and $l' = L(e_{i,t+1})$ for some trace $\mathcal{T}_i$ and time $t$.

$$\underset{\langle U, u_0, \delta_u, \delta_r \rangle}{\text{minimize}} \sum_{i \in I} \sum_{t \in T_i} \log(|N_{x_{i,t}, L(e_{i,t})}|) \tag{LRM}$$

$$s.t. \ \langle U, u_0, \delta_u, \delta_r \rangle \in \mathcal{R}_{\mathcal{P}} \tag{3}$$

$$|U| \leq u_{\max} \tag{4}$$

$$x_{i,t} \in U \qquad \forall i \in I, t \in T_i \cup \{t_i\} \tag{5}$$

$$x_{i,0} = u_0 \qquad \forall i \in I \tag{6}$$

$$x_{i,t+1} = \delta_u(x_{i,t}, L(e_{i,t+1})) \qquad \forall i \in I, t \in T_i \tag{7}$$

$$N_{u,l} \subseteq 2^{2^{\mathcal{P}}} \qquad\qquad\qquad \forall u \in U, l \in 2^{\mathcal{P}} \qquad (8)$$

$$L(e_{i,t+1}) \in N_{x_{i,t}, L(e_{i,t})} \qquad\qquad\qquad \forall i \in I, t \in T_i \qquad (9)$$

Constraints (3) and (4) ensure that we find a well-formed RM over $\mathcal{P}$ with at most $u_{\max}$ states. Constraint (5), (6), and (7) ensure that $x_{i,t}$ is equal to the current state of the RM, starting from $u_0$ and following $\delta_u$. Constraint (8) and (9) ensure that the sets $N_{u,l}$ contain every $L(e_{i,t+1})$ that has been seen right after $l$ and $u$ in $\mathcal{T}$. The objective function comes from maximizing the log-likelihood for predicting $L(e_{i,t+1})$ using a uniform distribution over all the possible options given by $N_{u,l}$.

A key property of this formulation is that any perfect RM is optimal w.r.t. the objective function in LRM when the number of traces tends to infinity (see supplementary material §3):

**Theorem 4.3.** *When the set of training traces (and their lengths) tends to infinity and is collected by a policy such that $\pi(a|o) > \epsilon$ for all $o \in O$ and $a \in A$, any perfect RM with respect to $L$ and at most $u_{max}$ states will be an optimal solution to the formulation LRM.*

Finally, note that the definition of a perfect RM does not impose conditions over the rewards associated with the RM (i.e., $\delta_r$). This is why $\delta_r$ is a free variable in the model LRM. However, we still expect $\delta_r$ to model the external reward signals given by the environment. To do so, we estimate $\delta_r(u, l)$ using its empirical expectation over $\mathcal{T}$ (as commonly done when constructing belief MDPs [5]).

### 4.3 Searching for a Perfect Reward Machine Using Tabu Search

We now describe the specific optimization technique used to solve LRM. We experimented with many discrete optimization approaches—including mixed integer programming [6], Benders decomposition [8], evolutionary algorithms [17], among others—and found local search algorithms [1] to be the most effective at finding high quality RMs given short time limits. In particular, we use Tabu search [11], a simple and versatile local search procedure with convergence guarantees and many successful applications in the literature [36]. We also include our unsuccessful mixed integer linear programming model for LRM in the supplementary material §1.

In the context of our work, Tabu search starts from a random RM and, on each iteration it evaluates all "neighbouring" RMs. We define the neighbourhood of an RM as the set of RMs that differ by exactly one transition (i.e., removing/adding a transition, or changing its value) and evaluate RMs using the objective function of LRM. When all neighbouring RMs are evaluated, the algorithm chooses the one with lowest values and sets it as the current RM. To avoid local minima, Tabu search maintains a *Tabu list* of all the RMs that were previously used as the current RM. Then, RMs in the Tabu list are pruned when examining the neighbourhood of the current RM.

## 5  Simultaneously Learning a Reward Machine and a Policy

We now describe our overall approach to simultaneously finding an RM and exploiting that RM to learn a policy. The complete pseudo-code can be found in the supplementary material (Algorithm 1).

Our approach starts by collecting a training set of traces $\mathcal{T}$ generated by a random policy during $t_w$ "warmup" steps. This set of traces is used to find an initial RM $\mathcal{R}$ using Tabu search. The algorithm then initializes policy $\pi$, sets the RM state to the initial state $u_0$, and sets the current label $l$ to the initial abstract observation $L(\emptyset, \emptyset, o)$. The standard RL learning loop is then followed: an action $a$ is selected following $\pi(o, u)$ where $u$ is the current RM state, and the agent receives the next observation $o'$ and the immediate reward $r$. The RM state is then updated to $u' = \delta_u(u, L(o, a, o'))$ and the last experience $(\langle o, u \rangle, a, r, \langle o', u' \rangle)$ is used by any RL method of choice to update $\pi$. Note that in an episodic task, the environment and RM are reset whenever a terminal state is reached.

If on any step, there is evidence that the current RM might not be the best one, our approach will attempt to find a new one. Recall that the RM $\mathcal{R}$ was selected using the cardinality of its prediction sets $N$ (LRM). Hence, if the current abstract observation $l'$ is not in $N_{u,l}$, adding the current trace to $\mathcal{T}$ will increase the size of $N_{u,l}$ for $\mathcal{R}$. As such, the cost of $\mathcal{R}$ will increase and it may no longer be the best RM. Thus, if $l' \notin N_{u,l}$, we add the current trace to $\mathcal{T}$ and search for a new RM. Recall that we use Tabu search, though any discrete optimization method could be applied. Our method only uses the new RM if its cost is lower than $\mathcal{R}$'s. If the RM is updated, a new policy is learned from scratch.

Given the current RM, we can use any RL algorithm to learn a policy $\pi(o, u)$, by treating the combination of $o$ and $u$ as the current state. If the RM is perfect, then the optimal policy $\pi^*(o, u)$ will

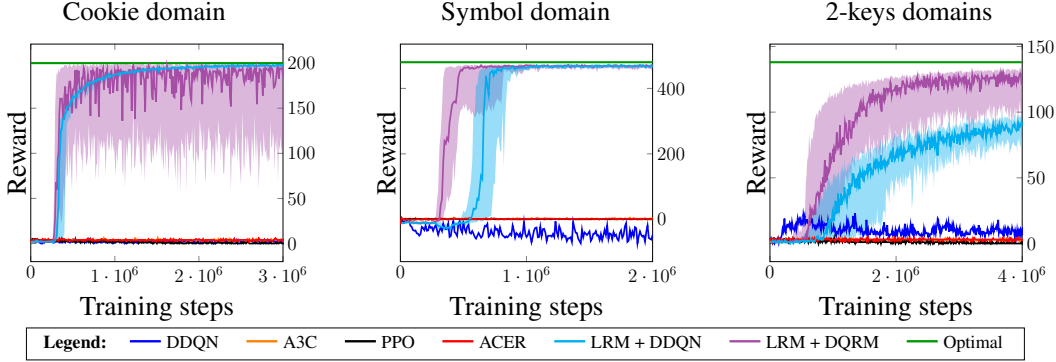

Figure 3: Total reward collected every $10,000$ training steps.

also be optimal for the original POMDP (as stated in Theorem 4.2). However, to exploit the problem structure exposed by the RM, we can use the QRM algorithm.

As explained in §3, standard QRM under partial observability can introduce a bias because an experience $e = (o, a, o')$ might be more or less likely depending on the RM state that the agent was in when the experience was collected. We partially address this issue by updating $\tilde{q}_u$ using $(o, a, o')$ if and only if $L(o, a, o') \in N_{u,l}$, where $l$ was the current abstract observation that generated the experience $(o, a, o')$. Hence, we do not transfer experiences from $u_i$ to $u_j$ if the current RM does not believe that $(o, a, o')$ is possible in $u_j$. For example, consider the cookie domain and the perfect RM from Figure 2c. If some experience consists of entering to the red room and seeing a cookie, then this experience will not be used by states $u_0$ and $u_3$ as it is impossible to observe a cookie at the red room from those states. Note that adding this rule may work in many cases, but it will not address the problem in all environments (more discussion in §7). We consider addressing this problem as an interesting area for future work.

## 6   Experimental Evaluation

We tested our approach on three partially observable grid domains (Figure 1). The agent can move in the four cardinal directions and can only see what is in the current room. These are stochastic domains where the outcome of an action randomly changes with a 5% probability.

The first environment is the *cookie domain* (Figure 1a) described in §3. Each episode is $5,000$ steps long, during which the agent should attempt to get as many cookies as possible.

The second environment is the *symbol domain* (Figure 1b). It has three symbols ♣, ♠, and ♦ in the red and blue rooms. One symbol from {♣, ♠, ♦} and possibly a right or left arrow are randomly placed at the yellow room. Intuitively, that symbol and arrow tell the agent where to go, e.g., ♣ and → tell the agent to go to ♣ in the east room. If there is no arrow, the agent can go to the target symbol in either room. An episode ends when the agent reaches any symbol in the red or blue room, at which point it receives a reward of $+1$ if it reached the correct symbol and $-1$ otherwise.

The third environment is the *2-keys domain* (Figure 1c). The agent receives a reward of $+1$ when it reaches the coffee (in the yellow room). To do so, it must open the two doors (shown in brown). Each door requires a different key to open it, and the agent can only carry one key at a time. Initially, the two keys are randomly located in either the blue room, the red room, or split between them.

We tested two versions of our Learned Reward Machine (LRM) approach: LRM+DDQN and LRM+DQRM. Both learn an RM from experience as described in §4.2, but LRM+DDQN learns a policy using DDQN [35] while LRM+DQRM uses the modified version of QRM described in §5. In all domains, we used $u_{\max} = 10$, $t_w = 200,000$, an epsilon greedy policy with $\epsilon = 0.1$, and a discount factor $\gamma = 0.9$. The size of the Tabu list and the number of steps that the Tabu search performs before returning the best RM found is $100$. We compared against 4 baselines: DDQN [35], A3C [24], ACER [37], and PPO [29] using the OpenAI baseline implementations [12]. DDQN uses the concatenation of the last 10 observations as input which gives DDQN a limited memory to better handle the domains. A3C, ACER, and PPO use an LSTM to summarize the history. Note that the

output of the labelling function was also given to the baselines. Details on the hyperparameters and networks can be found in the supplementary material §4.

Figure 3 shows the total cumulative rewards that each approach gets every $10,000$ training steps and compares it to the optimal policy. For the LRM algorithms, the figure shows the median performance over 30 runs per domain, and percentile 25 to 75 in the shadowed area. For the DDQN baseline, we show the maximum performance seen for each time period over 5 runs per problem. Similarly, we also show the maximum performance over the 30 runs of A3C, ACER, and PPO per period. All the baselines outperformed a random policy, but none make much progress on any of the domains.

Furthermore, LRM approaches largely outperform all the baselines, reaching close-to-optimal policies in the cookie and symbol domain. We also note that LRM+DQRM learns faster than LRM+DDQN, but is more unstable. In particular, LRM+DQRM converged to a considerably better policy than LRM+DDQN in the 2-keys domain. We believe this is due to QRM's experience sharing mechanism that allows for propagating sparse reward backwards faster (see supplementary material §4.3).

A key factor in the strong performance of the LRM approaches is that Tabu search finds high-quality RMs in less than 100 local search steps (Figure 5, supplementary material). In fact, our results show that Tabu search finds perfect RMs in most runs, in particular when tested over the symbol domain.

## 7 Discussion, Limitations, and Broader Potential

Solving partially observable RL problems is challenging and LRM was able to solve three problems that were conceptually simple but presented a major challenge to A3C, ACER, and PPO with LSTM-based memories. A key idea behind these results was to optimize over a necessary condition for perfect RMs. This objective favors RMs that are able to predict possible and impossible future observations at the abstract level given by the labelling function $L$. In this section, we discuss the advantages and current limitations of such an approach.

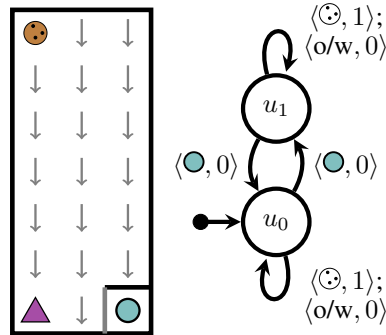

Figure 4: The gravity domain

We begin by considering the performance of Tabu search in our domains. Given a training set composed of one million transitions, a simple Python implementation of Tabu search takes less than 2.5 minutes to learn an RM across all our environments, when using 62 workers on a Threadripper 2990WX processor. Note that Tabu search's main bottleneck is evaluating the neighbourhood around the current RM solution. As the size of the neighbourhood depends on the size of the set of propositional symbols $\mathcal{P}$, exhaustively evaluating the neighbourhood may sometimes become impractical. To handle such problem, it will be necessary to import ideas from the Large Neighborhood Search literature [27].

Regarding limitations, learning the RM at the abstract level is efficient but requires ignoring (possibly relevant) low-level information. For instance, Figure 4 shows an adversarial example for LRM. The agent receives reward for eating the cookie ($\odot$). There is an external force pulling the agent down—i.e., the outcome of the "move-up" action is actually a downward movement with high probability. There is a button ($\bigcirc$) that the agent can press to turn off (or back on) the external force. Hence, the optimal policy is to press the button and then eat the cookie. Given $\mathcal{P} = \{\odot, \bigcirc\}$, a perfect RM for this environment is fairly simple (see Figure 4) but LRM might not find it. The reason is that pressing the button changes the low-level probabilities in the environment but does not change what is possible or impossible at the abstract level. In other words, while the LRM objective optimizes over necessary conditions for finding a perfect RM, those conditions are not sufficient to ensure that an optimal solution will be a perfect RM. In addition, if a perfect RM is found, our heuristic approach to share experiences in QRM would not work as intended because the experiences collected when the force is on (at $u_0$) would be used to update the policy for the case where the force is off (at $u_1$).

Other current limitations include that it is unclear how to handle noise over the high-level detectors $L$ and how to transfer learning from previously learned policies when a new RM is learned. Finally, defining a set of proper high-level detectors for a given environment might be a challenge to deploying LRM. Hence, looking for ways to automate that step is an important direction for future work.

# 8 Related Work

State-of-the-art approaches to partially observable RL use Recurrent Neural Networks (RNNs) as memory in combination with policy gradient [24, 37, 29, 15], or use external neural-based memories [25, 18, 13]. Other approaches include extensions to Model-Based Bayesian RL to work under partial observability [28, 7, 9] and to provide a small binary memory to the agent and a special set of actions to modify it [26]. While our experiments highlight the merits of our approach w.r.t. RNN-based approaches, we rely on ideas that are largely orthogonal. As such, we believe there is significant potential in mixing these approaches to get the benefit of memory at both the high- and the low-level.

The effectiveness of automata-based memory has long been recognized in the POMDP literature [5], where the objective is to find policies given a complete specification of the environment. The idea is to encode policies using Finite State Controllers (FSCs) which are FSMs where the transitions are defined in terms of low-level observations from the environment and each state in the FSM is associated with one primitive action. When interacting with the environment, the agent always selects the action associated with the current state in the controller. Meuleau et al. [22] adapted this idea to work in the RL setting by exploiting policy gradient to learn policies encoded as FSCs. RMs can be considered as a generalization of FSC as they allow for transitions using conditions over high-level events and associate complete policies (instead of just one primitive action) to each state. This allows our approach to easily leverage existing deep RL methods to learn policies from low-level inputs, such as images—which is not achievable by Meuleau et al. [22]. That said, further investigating using ideas for learning FSMs [3, 40, 10] in learning RMs is a promising direction for future work.

Our approach to learn RMs is greatly influenced by Predictive State Representations (PSRs) [20]. The idea behind PSRs is to find a set of core tests (i.e., sequences of actions and observations) such that if the agent can predict the probabilities of these occurring, given any history $H$, then those probabilities can be used to compute the probability of any other test given $H$. The insight is that state representations that are good for predicting the next observation are good for solving partially observable environments. We adapted this idea to the context of RM learning as discussed in §4.

While our work was under review, two interesting papers were submitted to arXiv. The first paper, by Xu et al. [39], proposes a polynomial time algorithm to learn reward machines in fully observable domains. Their goal is to learn the smallest reward machine that is consistent with the reward function—which makes sense for fully observable domains, but would have limited utility under partial observability (as discussed in §4). The second paper, by Zhang et al. [41], proposes to learn a discrete PSR representation of the environment directly from low-level observations and then plan over such representation using tabular Q-learning. This is a promising research direction, with some clear synergies with LRM.

# 9 Concluding Remarks

We have presented a method for learning (perfect) Reward Machines in partially observable environments and demonstrated the effectiveness of these learned RMs in tackling partially observable RL problems that are unsolvable by A3C, ACER and PPO. Informed by criteria from the POMDP, FSC, and PSR literature, we proposed a set of RM properties that support tackling RL in partially observable environments. We used these properties to formulate RM learning as a discrete optimization problem. We experimented with several optimization methods, finding Tabu search to be the most effective. We then combined this RM learning with policy learning for partially observable RL problems. Our combined approach outperformed a set of strong LSTM-based approaches on different domains.

We believe this work represents an important building block for creating RL agents that can solve cognitively challenging partially observable tasks. Not only did our approach solve problems that were unsolvable by A3C, ACER and PPO, but it did so in a relatively small number of training steps. RM learning provided the agent with memory, but more importantly the combination of RM learning and policy learning provided it with discrete reasoning capabilities that operated at a higher level of abstraction, while leveraging deep RL's ability to learn policies from low-level inputs. This work leaves open many interesting questions relating to abstraction, observability, and properties of the language over which RMs are constructed. We believe that addressing these questions, among many others, will push the boundary of partially observable RL problems that can be solved.

## Acknowledgments

We gratefully acknowledge funding from the Natural Sciences and Engineering Research Council of Canada (NSERC) and Microsoft Research. The first author also gratefully acknowledges funding from CONICYT (Becas Chile). A preliminary version of this work was presented at RLDM [34].

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
