[Supplementary Material]

# Learning Reward Machines for Partially Observable Reinforcement Learning

## Supplementary Material

## 1 Mixed Integer Linear Programming Model for `LRM`

In this section, we present a Mixed Integer Linear Programming model (`MILP`) for `LRM`. We assume $|U| = u_{\max}$ and set $K = 2^{2^{|\mathcal{P}|}}$. Variables $d_{u,u',l} \in \{0,1\}$ represent the possible transitions in the RM for each pair of states $u, u' \in U$ and abstract observation $l \in 2^{\mathcal{P}}$. Formally, this means that $d_{u,u',l} = 1$ iff $\delta_u(u,l) = u'$. Variable $w_{i,t,u} \in \{0,1\}$ indicates if the agent is at state $u \in U$ of the RM on trace $i \in I$ and time step $t \in T_i$, which corresponds to the statement that $w_{i,t,u} = 1$ iff $x_{i,t} = u$. Variable $p_{l,u,l'} \in \{0,1\}$ indicates if $l' \in 2^{\mathcal{P}}$ is a possible next abstract observation at RM state $u$ when observing $l \in 2^{\mathcal{P}}$. This means that $p_{l,u,l'} = 1$ iff $l' \in N_{u,l}$. Variable $y_{u,l,n} \in \{0,1\}$ represents the cardinality of $N_{u,l}$, meaning that $y_{u,l,n} = 1$ iff $|N_{u,l}| = n$. Lastly, variables $z_{i,t}$ represent the log-likelihood cost for trace $i \in I$ and time step $t \in T_i$, which can be formally stated as $z_{i,t} = \log(|N_{x_{i,t},L(e_{i,t})}|)$. The full model is then as follows:

$$\min \sum_{i \in I} \sum_{t \in T_i} z_{i,t} \qquad \text{(MILP)}$$

$$s.t.\ z_{i,t} \geq \sum_{n=1}^{K} y_{u,l,n} \cdot \log(n) - (1 - w_{i,t,u}) \cdot \log(K) \qquad \forall i \in I, t \in T_i, u \in U, l = L(e_{i,t}) \quad (10)$$

$$\sum_{n=1}^{K} y_{u,l,n} = 1 \qquad \forall u \in U, l \in 2^{\mathcal{P}} \quad (11)$$

$$\sum_{l' \in 2^{\mathcal{P}}} p_{l,u,l'} = \sum_{n=1}^{K} y_{u,l,n} \cdot n \qquad \forall u \in U, l \in 2^{\mathcal{P}}, n \in \{1..K\} \quad (12)$$

$$p_{l,u,l'} \geq w_{i,t,u} \qquad \forall i \in I, t \in T_i, l = L(e_{i,t}), l' = L(e_{i,t+1}) \quad (13)$$

$$\sum_{u' \in U} d_{u,u',l} = 1 \qquad \forall u \in U, l \in 2^{\mathcal{P}} \quad (14)$$

$$\sum_{u \in U} w_{i,t,u} = 1 \qquad \forall i \in I, t \in T_i, u \in U \quad (15)$$

$$w_{i,0,u_0} = 1 \qquad \forall i \in I \quad (16)$$

$$w_{i,t+1,u'} \geq w_{i,t,u} + d_{u,u',l} - 1 \qquad \forall i \in I, t \in T_i, u, u' \in U, l = L(e_{i,t+1}) \quad (17)$$

$$d_{u,u',l} \in \{0,1\} \qquad \forall u, u' \in U, l \in 2^{\mathcal{P}} \quad (18)$$

$$w_{i,t,u} \in \{0,1\} \qquad \forall i \in I, t \in T_i, u \in U \quad (19)$$

$$p_{l,u,l'} \in \{0,1\} \qquad \forall u \in U, l, l' \in 2^{\mathcal{P}} \quad (20)$$

$$y_{u,l,n} \in \{0,1\} \qquad \forall u \in U, l \in 2^{\mathcal{P}}, n \in \{1..K\} \quad (21)$$

$$z_{i,t} \geq 0 \qquad \forall i \in I, t \in T_i \quad (22)$$

Constraint (10) models the log-likelihood cost for each time step of a trace. Constraints (11) and (12) compute the cardinality of $N_{u,l}$. Constraint (13) defines the possible predictions given a trace. Constraint (14) enforces that for each RM state an abstract observation can lead to exactly one other RM state. Constraint (15) enforces that at any time step of a trace the agent can be at exactly one RM state. Constraint (16) imposes the initial RM state of a trace, and constraint (17) encodes the RM state transitions for a trace. Finally, constraints (18)-(22) correspond to the variables' domains.

## 2 Algorithm for Simultaneously Learning Reward Machines and a Policy

Algorithm 1 shows our overall approach to simultaneously learning an RM and exploiting that RM to learn a policy. The algorithm inputs are the set of propositional symbols $\mathcal{P}$, the labelling function $L$, a maximum on the number of RM states $u_{\max}$, and the number of "warmup" steps $t_w$. Our approach starts by collecting a training set of traces $\mathcal{T}$ generated by a random policy during $t_w$ steps (Line 2). This set of traces is used to find an initial RM $\mathcal{R}$ using Tabu search (Line 3). If later traces show that $\mathcal{R}$ is incorrect, our method will then find a new RM learned using the additional traces.

Lines 4 and 5 initialize the environment and the policy $\pi$, and set variables $x$ and $l$ to the initial RM state $u_0$ and initial abstract observation $L(\emptyset, \emptyset, o)$, respectively. Lines 7–19 are the main loop of our approach. Lines 7–10 are part of the standard RL loop: the agent executes an action $a$ selected following $\pi(o, x)$ and receives the next observation $o'$, the immediate reward $r$, and a boolean variable *done* indicating if the episode has terminated. Then, the state in the RM $x'$ is updated and the policy $\pi$ is improved using the last experience $(\langle o, x \rangle, a, r, \langle o', x' \rangle, done)$. Note that when *done* is true, the environment and RM are reset (Lines 17–18).

Lines 11–16 involve relearning the RM when there is evidence that the current RM might not be the best one. Recall that the RM $\mathcal{R}$ was selected using the cardinality of its prediction sets $N$ (see the description of LRM). Hence, if the current abstract observation $l'$ is not in $N_{x,l}$, then adding the current trace to $\mathcal{T}$ will increase the size of $N_{x,l}$ for $\mathcal{R}$. As such, the cost of $\mathcal{R}$ will increase and it may no longer be the best RM. Thus, if $l' \notin N_{x,l}$, we add the current trace to $\mathcal{T}$ and use Tabu search to find a new RM. Note, our method only uses the new RM if its cost is lower than that of $\mathcal{R}$ (Lines 14–16). However, when the RM is updated, a new policy is learned from scratch (Line 16).

---

**Algorithm 1** Learning an RM and a Policy

---

1: **Input:** $\mathcal{P}$, $L$, $A$, $u_{\max}$, $t_w$
2: $\mathcal{T} \leftarrow$ collect_traces($t_w$)
3: $\mathcal{R}, N \leftarrow$ learn_rm($\mathcal{P}, L, \mathcal{T}, u_{\max}$)
4: $o, x, l \leftarrow$ env_get_initial_state(), $u_0$, $L(\emptyset, \emptyset, o)$
5: $\pi \leftarrow$ initialize_policy()
6: **for** $t = 1$ **to** $t_{\text{train}}$ **do**
7:     $a \leftarrow$ select_action($\pi, o, x$)
8:     $o', r, \text{done} \leftarrow$ env_execute_action($a$)
9:     $x', l' \leftarrow \delta_u(x, L(o, a, o')), L(o, a, o')$
10:     $\pi \leftarrow$ improve($\pi, o, x, l, a, r, o', x', l'$, done, $N$)
11:     **if** $l' \notin N_{x,l}$ **then**
12:         $\mathcal{T} \leftarrow \mathcal{T} \cup$ get_current_trace()
13:         $\mathcal{R}', N \leftarrow$ relearn_rm($\mathcal{R}, \mathcal{P}, L, \mathcal{T}, u_{\max}$)
14:         **if** $\mathcal{R} \neq \mathcal{R}'$ **then**
15:             $\mathcal{R}, \text{done} \leftarrow \mathcal{R}'$, **true**
16:             $\pi \leftarrow$ initialize_policy()
17:         **end if**
18:     **end if**
19:     **if** done **then**
20:         $o', x', l' \leftarrow$ env_get_initial_state(), $u_0$, $L(\emptyset, \emptyset, o)$
21:     **end if**
22:     $o, x, l \leftarrow o', x', l'$
23: **end for**
24: **return** $\pi$

---

## 3    Theorems and Proof Sketches

**Theorem 4.1.** *Given any POMDP $\mathcal{P}_{\mathcal{O}}$ with a finite reachable belief space, there exist a perfect RM for $\mathcal{P}_{\mathcal{O}}$ with respect to some labelling function L.*

*Proof sketch.* If the reachable belief space $B$ is finite, we can construct an RM that keeps track of the current belief state using one RM state per belief state and emulating their progression using $\delta_u$, and one propositional symbol for every action-observation pair. Thus, the current belief state $b_t$ can be inferred from the last observation, last action, and the current RM state. As such, the equality from Definition 4.1 holds.    □

Two interesting properties follow from the definition of a perfect RM. First, if the set of belief states $B$ for the POMDP $\mathcal{P}_{\mathcal{O}}$ is finite, then there exists a perfect RM for $\mathcal{P}_{\mathcal{O}}$ with respect to some $L$. Second, the optimal policies for perfect RMs are also optimal for the POMDP.

**Theorem 4.2.** *Let $\mathcal{R}_{\mathcal{P}}$ be a perfect RM for a POMDP $\mathcal{P}_{\mathcal{O}}$ w.r.t. a labelling function L, then any optimal policy for $\mathcal{R}_{\mathcal{P}}$ w.r.t. the environmental reward is also optimal for $\mathcal{P}_{\mathcal{O}}$.*

*Proof sketch.* As the next observation and immediate reward probabilities can be predicted from $O \times U \times A$, an optimal policy over $O \times U$ must also be optimal over $\mathcal{P}_{\mathcal{O}}$.    □

A key property of this formulation is that any perfect RM is optimal with respect to the objective function in LRM when the number of traces tends to infinity:

**Theorem 4.3.** *When the set of training traces (and their lengths) tends to infinity and is collected by a policy such that $\pi(a|o) > \epsilon$ for all $o \in O$ and $a \in A$, and some constant $\epsilon > 0$, then any perfect RM with respect to L and at most $u_{max}$ states will be an optimal solution to the formulation given in* LRM.

*Proof sketch.* In the limit, $l' \in N_{u,l}$ if and only if the probability of observing $l'$ after executing an action from the RM state $u$ while observing $l$ is non-zero. In particular, for all $i \in I$ and $t \in T$, the cardinality of $N_{x_{i,t},L(e_{i,t})}$ will be minimal for a perfect RM. This follows from the fact that perfect RMs make perfect predictions for the next observation $o'$ given $o$, $u$, and $a$. Therefore, as we minimize the sum over $\log(|N_{x_{i,t},L(e_{i,t})}|)$, the objective value for a perfect RM must be minimal.    □

# 4  Experimental Evaluation

## 4.1  Experimental Details

For LRM+DDQN and LRM+DQRM, the neural network used has 5 fully connected layers with 64 neurons per layer. On every step, we trained the network using 32 sampled experiences from a replay buffer of size 100,000 using the Adam optimizer [19] and a learning rate of 5e-5. The target networks were updated every 100 steps.

DDQN [35] uses the same parameters and network architecture as LRM+DDQN, but its input is the concatenation of the last 10 observations, as commonly done by Atari playing agents. This gives DDQN a limited memory to better handle partially observable domains. In contrast, A3C, ACER, and PPO use an LSTM to summarize the history.

We also followed the same testing methodology that was used in their original publications. We ran each approach at least 30 times per domain, and on every run, we randomly selected the number of hidden neurons for the LSTM from $\{64, 128, 256, 512\}$ and a learning rate from $(1e\text{-}3, 1e\text{-}5)$. We also sampled $\delta$ from $\{0, 1, 2\}$ for ACER and the clip range from $(0.1, 0.3)$ for PPO. Other parameters were fixed to their default values.

While interacting with the environment, the agents were given a "top-down" view of the world represented as a set of binary matrices. One matrix had a 1 in the current location of the agent, one had a 1 in only those locations that are currently observable, and the remaining matrices each corresponded to an object in the environment and had a 1 at only those locations that were both currently observable and contained that object (i.e., locations in other rooms are "blacked out"). The agent also had access to features indicating if they were carrying a key, which colour room they were in, and the current status (i.e., occurring or not occurring) of the events detected by the labelling function.

## 4.2  Tabu Search

Figure 5 evaluates the quality of the RMs found by Tabu search by comparing it the perfect RM. In each plot, a dot compares the cost of a handcrafted perfect RM with that of an RM $\mathcal{R}$ that was found by Tabu search while running our LRM approaches, where the costs are evaluated relative to the training set used to find $\mathcal{R}$. Being on or under the diagonal line (as in most of the points in the figure) means that Tabu search is finding RMs whose values are at least as good as the handcrafted RM. Hence, Tabu search is either finding perfect RMs or discovering that our training set is incomplete and our agent will eventually fill those gaps.

Figure 5: Cost comparison between perfect RM and RM found by Tabu search.

## 4.3  DDQN vs DQRM: Exploration Heatmaps and Learned Trajectories

As shown in §6 of the paper, LRM+DQRM tends to learn faster than LRM+DDQN and largely outperforms LRM+DDQN in the 2-keys domain. Towards better understating these results, we ran the following experiment. For the 2-keys domain (and identical random seed), we learn policies using DDQN and DQRM over the same handcrafted perfect RM from Figure 6.

In the 2-keys domain, $\mathcal{P} = \{🔑, 🔑🔑, \blacktriangle, ☕, 🟥, ⬜, 🟦, 🟨\}$. These properties are true in the following situations: 🟥, ⬜, 🟦, or 🟨 is true if the agent ends the experience in a room of that color; 🔑 is true if the agent ends the experience in the same room as exactly one key; 🔑🔑 is true if the agent ends the experience in the same room as exactly two keys; $\blacktriangle$ is true if the agent is carrying a key; and ☕ is true if the agent reaches the coffee with its last action.

Figure 6: A perfect reward machine for the 2-keys domain.

Figures 7 and 8 show the exploration heatmaps of DDQN and DQRM agents during the first $500,000$ training steps (red areas represent places where the agent spent more time). For both approaches, the first $50,000$ steps are random. After that, learning begins and both agents follow an $\epsilon$-greedy policy with $\epsilon = 0.1$. The most interesting difference between DDQN and DQRM is between training steps $50,000$ and $150,000$ (second and third maps from the left in Figure 7). They show that DQRM spends less time in the coffee room and the hallway than DDQN. This could be explained by the two main differences between these RL methods, as detailed below.

Figure 7: Exploration heatmaps for DDQN and DQRM over the 2-keys domain given a perfect RM.

First, DQRM decomposes the main policy into one q-network per state in the RM. Hence, each q-network has to learn a relatively simpler policy than DDQN. In fact, the q-network for the last

RM state $u_6$ is fairly simple: when the agent is carrying a key and there is only one closed door, go and get the coffee. As expected, DQRM seems to learn an accurate estimation of that q-function using very limited interactions within the coffee room. In contrast, DDQN uses one big q-network to model the complete policy. Receiving reward by getting the coffee pushes the q-network estimation to believe that a high reward can be collected from the coffee room (even if the doors are closed). Hence, the agent spent a considerable amount of time hitting the second door without having a key.

Second, DQRM shares experience over all the q-networks. This allows for using the experiences collected while being at early stages of the task (e.g., states $u_0$, $u_1$, $u_2$, and $u_3$) to update policies for later stages (e.g., states $u_4$, $u_5$, and $u_6$). In particular, all the experience needed to learn to navigate from one room to another is shared. Therefore, while DDQN would depend on its network to avoid relearning how to navigate between rooms when the RM state changes, DQRM enforces such transfer by sharing experiences whenever it is appropriate. This might explain why DDQN spends considerably more time in the hallway than DQRM in heatmaps 50-100K and 100-150K.

| DDQN 250–300K steps | DDQN 300–350K steps | DDQN 350–400K steps | DDQN 400–450K steps | DDQN 450–500K steps |
|---|---|---|---|---|

| DQRM 250–300K steps | DQRM 300–350K steps | DQRM 350–400K steps | DQRM 400–450K steps | DQRM 450–500K steps |
|---|---|---|---|---|

Figure 8: Exploration heatmaps for DDQN and DQRM over the 2-keys domain given a perfect RM.

Note that the exploratory trend from 150-500K follows a clear pattern. On one hand, the DQRM agent seems to have a good idea of how to solve the task and, therefore, it spends most of its time on the hallway (solving the tasks requires passing through the hallway at least 4 times). On the other hand, the DDQN agent is getting stuck exploring subregions of the map.

Finally, we inspected the trajectories of each agent when solving the task after 1 million training steps. Figures 9 shows the DDQN agent and Figure 10 shows the DQRM agent. Both agents solved the task, but DQRM solved it faster (83 steps vs 102 steps). In solving this problem, the main difficulty for the DDQN agent is in reacting to entering the South room and discovering that the keys are not there. Its reaction is to enter and leave the empty room many times before going to the North room. In the case of DQRM, the agent goes directly to the North room after observing that the South room is empty, but then it enters and leaves the North room a few times before collecting the keys. After collecting the first key, both agents solved the rest of the task almost optimally.

(a) Collecting the first key.  (b) Opening the first door.

(c) Collecting the other key.  (d) Opening the second door.

Figure 9: The learned trace after 1 million training steps by DDQN given a perfect RM, divided into four stages.

(a) Collecting the first key.  (b) Opening the first door.

(c) Collecting the other key.  (d) Opening the second door.

Figure 10: The learned trace after 1 million training steps by QRM given a perfect RM, divided into four stages.