[Reviews · NeurIPS 2019]

Reviewer 1



The authors propose a novel approach for solving POMDPs by simultaneously learning and solving reward machines. The method relies on building a finite state machine which properly predicts possible observations and rewards. The authors demonstrate that their method outperforms baselines in three different partially observable gridworlds. Overall, I found the paper clear and well motivated. Learning to solve POMDPs is a very challenging problem and any progress or insight has the potential to have a big impact. My main concern about this work concerns the scalability of the discrete optimization problem. The experimental results don’t provide much insight in that regards. How long does solving the LRM take in these gridworlds? How often are the LRM recomputed in practice? While better learning curves do confirm that the algorithm works, they don’t usually provide much insight about how algorithm behaves. Additionally, without more information about the domain, it is difficult to appreciate the performance difference. For instance, without knowing how many steps are required to traverse the rooms, I can’t know if the 10 steps of history given to DDQN is a too much, enough or too little. I would expect that a sufficiently long history would enable DDQN to solve the task. Have the authors tried this? While I am disappointed by the empirical evaluation, the conceptual and theoretical contributions has me leaning towards acceptance. There is a considerable amount of work that focuses on improving discrete and hybrid optimization, and being able to make use of that is very appealing. I found the decision to replace the PSR’s optimization criterion (i.e., Markovian predictions) with the more limited constraints on possible observations quite interesting. Do the authors have any intuition about when this is not a good proxy? When would this lead to less desirable RMs? How does this work relate to [1] which also follows the PSR approach? Additionally, how does this scale when there are a large number of possible observations for some/all reward machine states? Misc comments: - Line 81, typo “[...] only see what _it_ is in the room that [...]” - Experiments, how many steps does it take to go from one room to another? - Line 294, 295, “the maximum reward”, is it meant to say the maximum cumulative rewards/return? - Why not evaluate each method the same way? This seems unnecessary and a bit odd. [1] Zhang, Amy, et al. "Learning Causal State Representations of Partially Observable Environments." arXiv preprint arXiv:1906.10437 (2019). Post-rebuttal: ----------------------- My concerns have been addressed and I recommend acceptance.

Reviewer 2



# Originality The work presents a well thought-out extension of RMs to POMDPs, and a learning algorithm to jointly learn them with state of the art model-free methods. The manuscript contains a sufficient literature review, and makes it clear about the distinctions and improvements from previous work. # Quality The overall quality of the work is high. The presented framework and learning method seem theoretical sound (at least under a quick scrutiny), and the experimental results are sound - albeit tiny in content and, possibly, analysis. That said, the authors are careful about their claims, and seem pretty forthcoming regarding the weaknesses of LRM. # Clarity The paper is very well written, and mostly clear w.r.t. its notation and presentation. The manuscript and its supplementary materials are dense of both theoretical and implementation details. I feel relatively confident myself and most other RL researchers could have a good attempt at reproducing the method and the experimental settings purely based on the writeup, however I hope that the authors will nonetheless release the code as declared in the manuscript. # Significance Reward Machines have yet to become part of the common set of tools used by RL researchers and practitioners, primarily because of how recent they are, and the relatively simple settings that have been used to evaluate them so far. This work doesn't fix this particular issue, however it demonstrates that for a class of common environment settings - i.e. partially observable, sparse-reward - they have the potential of being a good way to incorporate priors about the task structure into policy learning and inference, which is an extremely important topic of research in the RL community.

Reviewer 3



The authors apply Reward Machines (RMs) [26] to POMDP problems. While RMs is a automate-based representation of the reward function and originally is proposed for decomposing problems into subproblems, it is used for a kind of belief state representation. While RMs were given in the previous work, the paper proposes a method of learning RMs from data. The task as learning RMs is formulated as a discrete optimization problem. The authors also show the effectiveness of the proposed method with numerical experiments. The proposed method had much better performance than A3c, ACER, and PPO with LSTM. However, I have the following concerns. Concerns: [A] It appears to be non-trivial to prepare the labeling function L. It would be useful to give examples of the labeling function in several POMDP tasks. [B] Numerical experiments are weak. It would be required to experiment with various POMDP benchmark tasks or practical tasks. The baseline methods seem to be biased. I think that the proposed method can be regarded as a kind of the model-based approach for POMDP. So model-based method for POMDP like [*] should be included in baseline methods. [C] I could not follow Proof sketch of Theorem 4.3, especially the last sentence. I feel that there is some discrepancy between Definition 4.1 for the perfect RM and the setting of the objective value in (LRM), which is the sum over log(|N_*|). Minor issue: * In page 13, Theorem 3.* should be Theorem 4.*? [*] Doshi-Velez, Finale P.; Pfau, David; Wood, Frank; Roy, Nicholas. Bayesian Nonparametric Methods for Partially-Observable Reinforcement Learning, IEEE Transactions on Pattern Analysis and Machine Intelligence, Volume: 37, Issue: 2 , 2015.

[Author Response · NeurIPS 2019]

Thank you for your excellent feedback. Solving Partially Observable RL is challenging
and, as noted by R3, the proposed method had much better performance than A3C, ACER,
and PPO with LSTM. We address your main questions here and will elaborate in the paper.

Figure 1: Gravity domain

**How well does Tabu search scale? (R1)**: Roughly speaking, Tabu search scales as well as
genetic algorithms (which are commonly used in RL). Its main bottleneck is evaluating the
neighbourhood around the current RM solution, but this step is easily parallelizable. Given
a training set composed of 1 million transitions, a simple Python implementation of Tabu
search took less than 2.5 minutes to learn an RM across all our environments (using 62 workers). In our experiments, the
agent relearned the RM 8.6 times (on average) per run. Note that the size of the neighbourhood depends on the number
of possible abstract observations, and so exhaustively evaluating the neighbourhood may sometimes become impractical.
This well-studied problem has plenty of proposed solutions (known as Large Neighborhood Search methods), though.

**What are the limitations of LRM (R1) and when might QRM not work as intended (R2)?**: As R1 mentioned, an
interesting idea from LRM was to optimize over a necessary condition for perfect RMs. This objective favors RMs that
are able to predict possible and impossible future observations at the abstract level given by the labelling function $L$.
Learning the RM at the abstract level is efficient but requires ignoring (possibly relevant) low-level information. (In the
future, we would like to learn LSTM policies inside the RM to account for any missing information in $L$.)

To the limitations, Figure 1 shows an adversarial example for LRM. The agent receives reward for eating the cookie ($\odot$).
There is an external force pulling the agent down—i.e., the outcome of the "move up" action is actually a downward
movement with high probability. There is a button ($\bullet$) that the agent can press to turn off (or back on) the external force.
Hence, the optimal policy is to press the button and then eat the cookie. Given $\mathcal{P} = \{\odot, \bullet\}$, a perfect RM for this
environment is fairly simple (see Figure 1) but LRM might not find it. The reason is that pressing the button changes
the low-level probabilities in the environment but does not change what is possible or impossible at the abstract level.
Moreover, if a perfect RM is found, our heuristic approach to share experiences in QRM would not work as intended
because the experiences collected when the force is on (at $u_0$) would be used to update the policy for the case where the
force is off (at $u_1$). From a practical perspective, a simple solution is to add a high-level detector that senses the external
force. Nonetheless, we will discuss the theoretical implications of this interesting adversarial example in the paper.

**Question about the evaluation metric and grid sizes (R1)**: Indeed, we are reporting the maximum "*cumulative*"
rewards for the baselines every $10,000$ steps. The idea was to highlight that no run of the baselines learned to
consistently solve the tasks. We used small grids in our experiments (it takes 8 steps to go from one room to another).
Hence, a 10-order memory for DDQN should be enough (in principle) to learn to solve our environments.

**Relation with Zhang et al.'s work (R1)**: This interesting work—which was submitted to arxiv after the NeurIPS
deadline—will definitely be discussed in our paper. In short, the main distinction is that LRM exploits a high-level
abstraction of the observations to understand how to decompose the problem into Markovian subproblems. Exploiting
abstractions has historically helped RL agents to solve challenging domains. In contrast, Zhang et al.'s work has the
merit of learning PSRs at the low-level, making it easier to use as an off-the-shelf tool for Partially Observable RL.

**What if the high-level detectors were noisy? (R2)**: To handle noise over $L$ (without requiring an explosion of the
size of the RM), it seems necessary to move from deterministic to stochastic RMs. This is to allow the agent to be at
multiple RM states with certain probability. We believe this is an interesting research direction.

**Are you planning to release your code? Does LRM+DDQN/DQRN require a stochastic environment? Could
you add more empirical evaluation on why LRM+DQRM converges faster than LRM+DDQN? (R2)**: We will
release our code. Our approach works well on the deterministic version of our environments too. We will add further
information, including exploration heatmaps and learned trajectories, to the supplementary materials.

**How to prepare an effective labelling function $L$? (R3)**: Intuitively, any event that might be useful for the agent to
remember is a good candidate to be included in $L$. We are also interested in investigating methods for learning a suitable
$L$ during environment interaction, and feel this paper demonstrates how one can be exploited if learned (or given).

**Is LRM a model-based RL approach and, as such, shouldn't it be compared with Doshi-Velez et al.? (R3)**:
Thanks for pointing this out. LRM+DDQN/DQRM lies somewhere between model-based and model-free as the RM is
learned in a model-based fashion but its policies are learned in a model-free way. This allows our models to leverage
deep RL's ability to learn policies from low-level inputs (e.g., images). As such, our baselines and part of the related
work discussion was indeed biased towards deep RL approaches for Partially Observable RL. We will partially remedy
this by including a discussion about non-Parametric methods, including Doshi-Velez et al., in the related work section.

**Clarification in proof sketch of Theorem 4.3 (R3)**: The discrepancy between Def 4.1 and the LRM's objective value
comes from the fact that LRM is optimizing over a necessary (but not sufficient) condition for finding a perfect RM. If
this does not answer your question, please let us know and we will further elaborate on a revised version of this work.

[Meta-Review · NeurIPS 2019]

All the reviewers recommended acceptance.